# Automatized Detection of Periodontal Bone Loss on Periapical Radiographs by Vision Transformer Networks

**DOI:** 10.3390/diagnostics13233562

**Published:** 2023-11-29

**Authors:** Helena Dujic, Ole Meyer, Patrick Hoss, Uta Christine Wölfle, Annika Wülk, Theresa Meusburger, Leon Meier, Volker Gruhn, Marc Hesenius, Reinhard Hickel, Jan Kühnisch

**Affiliations:** 1Department of Conservative Dentistry and Periodontology, LMU University Hospital, LMU Munich, 80336 Munich, Germany; patrick.hoss@t-online.de (P.H.); uwoelfle@dent.med.uni-muenchen.de (U.C.W.); annika.wuelk@gmx.de (A.W.); theresa.meusburger@hotmail.com (T.M.); meierl.leon@gmail.com (L.M.); hickel@dent.med.uni-muenchen.de (R.H.); 2Institute for Software Engineering, University of Duisburg-Essen, 45127 Essen, Germany; ole.meyer@uni-due.de (O.M.); volker.gruhn@paluno.uni-due.de (V.G.); marc.hesenius@uni-due.de (M.H.)

**Keywords:** artificial intelligence, deep learning, machine learning, transformer, periapical radiographs, periodontitis, periodontal bone loss, diagnostics

## Abstract

Several artificial intelligence-based models have been presented for the detection of periodontal bone loss (PBL), mostly using convolutional neural networks, which are the state of the art in deep learning. Given the emerging breakthrough of transformer networks in computer vision, we aimed to evaluate various models for automatized PBL detection. An image data set of 21,819 anonymized periapical radiographs from the upper/lower and anterior/posterior regions was assessed by calibrated dentists according to PBL. Five vision transformer networks (ViT-base/ViT-large from Google, BEiT-base/BEiT-large from Microsoft, DeiT-base from Facebook/Meta) were utilized and evaluated. Accuracy (ACC), sensitivity (SE), specificity (SP), positive/negative predictive value (PPV/NPV) and area under the ROC curve (AUC) were statistically determined. The overall diagnostic ACC and AUC values ranged from 83.4 to 85.2% and 0.899 to 0.918 for all evaluated transformer networks, respectively. Differences in diagnostic performance were evident for lower (ACC 94.1–96.7%; AUC 0.944–0.970) and upper anterior (86.7–90.2%; 0.948–0.958) and lower (85.6–87.2%; 0.913–0.937) and upper posterior teeth (78.1–81.0%; 0.851–0.875). In this study, only minor differences among the tested networks were detected for PBL detection. To increase the diagnostic performance and to support the clinical use of such networks, further optimisations with larger and manually annotated image data sets are needed.

## 1. Introduction

Periodontitis is a chronic inflammatory disease of the supporting dental tissues and affects a relevant proportion of the world’s population [1,2,3,4]. Furthermore, periodontitis can also be associated with various risk factors such as smoking and stress, as well as systemic diseases such as diabetes mellitus or pulmonary diseases. Clinically, periodontitis is associated with periodontal bone loss (PBL), tooth loosening and tooth loss. All of these factors can further impair functionality, aesthetics and quality of life [5,6]. Considering the recommendations of the latest workshop on the classification of periodontal diseases [7,8], the initial diagnosis is primarily based on clinical assessment, bleeding on probing, repeated measurements of clinical attachment loss and probing pocket depth. The early manifestations of periodontitis are only clinically recognisable. Furthermore, staging based on the radiographic assessment of PBL is considered possible only with the progression of the disease. As a result, the importance of radiographs increases as the disease progresses, since the extent of alveolar bone changes can be visualized more accurately [9,10]. However, a reliable assessment of PBL remains susceptible to diagnostic subjectivity among dentists [11,12]. Therefore, the use of image analysis tools based on artificial intelligence (AI) methods could possibly enable the automated assessment of PBL on radiographs and potentially improve diagnostic accuracy. Interestingly, several research groups have developed AI-based algorithms and published promising results on panoramic [11,13,14,15,16,17,18,19,20,21] and periapical radiographs [12,22,23,24,25,26,27,28,29,30]. Looking at the methodology of the studies published so far, almost all research groups have used an image set of a limited size to train different types of convolutional neural networks (CNNs). This has led to heterogeneous but promising results [31,32]. In particular, more than half of the studies published to date have reported a data set of less than 1000 X-ray images [12,14,15,16,17,18,19,21,26,29,30]. In addition, some studies used different exclusion criteria for their data set, meaning that radiographs with a specific tooth group or radiographs with caries or root canal treatment were excluded (e.g., [23,28]). In addition, variability in the architecture of the CNNs used can be observed, e.g., ResNet, U-Net and faster R-CNNs were trained for PBL detection [12,13,15,17,18,19,25]. Accurate manual annotation also contributed significantly to the reported results, as studies reporting the annotation of radiologic features of PBL described a better diagnostic performance, e.g., [13,25]. Moreover, none of the previously mentioned studies used recently introduced transformer networks for computer vision tasks, which are the most recent available technology for automatized image analysis and may possibly outperform current CNNs in the future [33]. On the one hand, CNNs have proven their value in tasks such as image classification and segmentation by efficiently processing large data sets. Among the most significant advantages is the ability of CNNs to recognize local patterns, such as edges or shapes. This proved to be particularly helpful for recognizing features in dental X-rays, such as tooth decay, different tooth shapes, etc. On the other hand, the vision transformer’s attention mechanism allows the model to learn the correlation of parts of the image that may not be in direct proximity. In the case of PBL detection, these are primarily the cementoenamel junction, alveolar bone and apex, as well as other anatomical structures relevant for the evaluation. Notably, transformer networks usually require a larger amount of training data compared to CNNs. Following this, we aimed to compare the diagnostic performance of five different transformer networks for automatized PBL detection on periapical radiographs. Specifically, it was hypothesized that the diagnostic performance of the included transformer networks would be similar and that an overall diagnostic accuracy of 90% would be achievable.

## 2. Materials and Methods

### 2.1. Study Design

The Ethics Committee of the Medical Faculty of Ludwig Maximilian University (LMU) of Munich approved this study protocol (project number 020-798). The periapical radiographs used in this study were anonymized and obtained as part of previous clinical examinations. Consequently, we could not identify any of the patients and were therefore unable to obtain written informed consent. The reporting of this research followed the Standard for Reporting of Diagnostic Accuracy Studies (STARD) Steering Committee recommendations [34] as well as the recommendations for reporting AI studies in dentistry [35].

### 2.2. Periapical Radiographs

This study used anonymized periapical radiographs (Figure 1). All X-rays were taken at the Department of Conservative Dentistry and Periodontology (LMU University Hospital) and different dental practices. To ensure a high-quality image sample, exclusion criteria were previously defined. This involved excluding distorted radiographs, radiographs with overlapping teeth, radiographs with artifacts, and radiographs with incompletely imaged teeth for which an assessment of the periodontium was not possible. Furthermore, radiographs with implants, with endodontic treatments or photographed radiographs, were also excluded. Further exclusion criteria were not defined. All periapical radiographs were stored in .jpg format and processed without downsizing the original resolution. Altogether, 21,819 periapical radiographs, divided into upper/lower anterior and posterior teeth, were selected for this study (Table 1). The majority of the radiographs were upper (*N* = 9461) and lower posterior teeth (*N* = 8425), outnumbering upper (*N* = 1944) and lower anterior teeth (*N* = 1989). Additionally, the radiographs were categorized according to PBL.

### 2.3. Categorisation of Periodontal Bone Loss (Reference Standard)

All radiographs were precategorised by a group of graduate dentists (P.H., T.M., A.W. and L.M.) and later independently counterchecked by experienced examiners (H.D., U.W. and J.K.). For each of the periapical radiographs, a diagnosis was made by differentiating between healthy teeth and teeth affected by mild, moderate or severe PBL [7,8]. Clinical data were not available prior to decision making. In detail, the following diagnostic criteria were applied: 0—radiographic PBL not detectable; 1—mild radiographic PBL up to 15% of the root length; 2—moderate radiographic PBL between 15% and 33% of the root length; and 3—severe radiographic PBL extending to the mid-third of the root and beyond (Figure 1). In the case of divergent opinions, each radiograph was discussed until consensus was reached. Each dichotomized diagnostic decision (0 versus 1 to 3)—one per image—served as a reference standard for the cyclic training and repeated evaluation of the deep learning-based transformer network.

Before conducting this study, all participating dentists were trained during a 2-day workshop by the principal investigator (J.K.). Following this workshop, the effectiveness of training was determined during a calibration course. The inter- and intra-examiner reproducibility for PBL were assessed on 150 periapical radiographs. The corresponding Kappa values showed substantial reliability, ranging from 0.454 to 0.482 (inter-examiner). The intra-examiner reliability in terms of Cohen’s Kappa amounted to 0.739 [36].

### 2.4. Training of the Deep Learning-Based Transformer Networks (Test Method)

A pipeline of well-established methods was used to train the transformer networks. In principle, the entire image set of 21,819 periapical radiographs was divided into a training set (*N* = 18,819) and a test set. The latter included 3000 randomly selected X-rays from the overall image set and served as an independent test set that was not included in the model training. Given the high number of periapical radiographs in our data set, image augmentation and preprocessing were not necessary. Furthermore, all X-rays had a standardized size.

The previously mentioned data set was used to train five different pre-trained transformer networks (Table 2) [33,37,38]. The learning performance was evaluated with the independent test set. The used transformer networks were trained by using backpropagation to determine the gradient for learning. Furthermore, the model training was accelerated by the use of Floating Point 16 and a university-based computer (i9 10850K 10 × 3.60 GHz, Intel Corp., Santa Clara, CA, USA) equipped with 64 GB RAM and a professional graphic card (RTX A6000 48 GB (Nvidia, Santa Clara, CA, USA). The batch size amounted to 16 randomly selected images. Each transformer was trained over 5 epochs with cross entropy loss as an error function and an application of the Adam optimizer (Betas 0.9 and 0.999, Epsilon × 10^−8^).

### 2.5. Statistical Analysis

The data were analysed using Python (version 3.8.5, http://www.python.org accessed on 28 November 2023). The diagnostic ACC was determined by calculating the number of true negatives (TN), true positives (TP), false positives (FP) and false negatives (FN). In addition, the sensitivity (SE), specificity (SP), positive/negative predictive values (PPV/NPV) and area under the receiver operating characteristic (ROC) curve were calculated [39].

## 3. Results

In the present study, we calculated the diagnostic performance for automatized PBL detection on periapical radiographs for lower/upper and anterior/posterior teeth altogether (Table 3) and separately (Table 4) by using five different transformer networks. In general, when analysing the whole data set of periapical radiographs, the ACC ranged from 83.4% to 85.2%; the corresponding AUC values ranged from 0.899 to 0.918 (Figure 2). The detailed data analysis revealed generally better performance data for mandibular teeth than for maxillary teeth (Table 4). Here, the ACC ranged from 94.1% to 96.7% for mandibular anteriors and from 85.6% to 87.2% for mandibular posteriors. The corresponding data for maxillary anterior and posterior teeth varied between 86.7% and 90.2% as well as between 78.1% and 81.0%, respectively. Additionally, the AUC values tended to be similar or better for mandibular teeth (Table 4). Furthermore, the SE values were consistently higher than the SP values.

When comparing the metrics of the included transformer networks, only minor differences appeared in the results (Table 3 and Table 4). However, the ACC and AUC values were found to be high in all scenarios, and SE was higher than SP.

## 4. Discussion

The present study aimed to compare the diagnostic performance of five different transformer networks for automatized PBL detection on periapical radiographs. Depending on the applied network, the overall diagnostic ACC and AUC values ranged from 83.4% to 85.2% and 0.899 to 0.918, respectively (Table 3, Figure 2). On the one hand, the ACC values must be evaluated as high; on the other hand, the hypothesized overall diagnostic ACC of 90% was not achieved. Therefore, the initially formulated hypothesis must be rejected.

When comparing the documented diagnostic performance data (Table 3 and Table 4) with data from the literature, the following conclusion can be drawn. In general, the majority of comparable studies presented model performances in the same or lower order of magnitude [11,12,13,14,15,17,20,21,23,26,28,40], whereas only a few studies registered above-average values [25,41]. In detail, Lee et al. [25] reported an ACC for staging that ranged from 88% to 99%. They further stated that the ACC for periodontitis case classification was 85%. Specifically, 693 periapical radiographs were independently annotated by examiners prior to training the model, indicating regions of interest such as the alveolar bone, presence of teeth, cementoenamel junctions and presence of restorations. In addition, a further 644 periapical radiographs were used to assess the ACC of the model. In another study on staging, Widyaningrum et al. [41] stated that the detection rate was 95%, with the best performance shown for stage 4 periodontitis. Although the data set consisted of only 100 panoramic radiographs, two investigators annotated the previously mentioned radiographs before training the CNN. Accurate annotations were made by marking the alveolar ridge and the alveolar bone surrounding the teeth. In addition, the examiners added a number indicating the stage of periodontitis. Therefore, the few studies with better diagnostic performance seem remarkable compared to other studies with results of a lower magnitude. Here, other dental detection tasks should also be mentioned in comparison, where a higher ACC—typically approximately 90%—was usually registered with a similar methodology, e.g., in the detection of caries or periapical lesions on radiographs (e.g., [42,43,44]) and the detection of clinical pathologies or restorations on intraoral photographs (e.g., [45,46,47,48,49]). This may indicate that automatized PBL detection is more difficult to accomplish, which is supported by the fact that PBL characteristics are usually spread over the whole radiographic image and can have varying extents.

Our study revealed differences in the performance of the model in relation to the analysed group of teeth. In principle, automatized PBL detection performed better for mandibular teeth than for maxillary teeth, and better for anterior teeth compared to posterior teeth (Table 4). Only a few studies have considered this aspect thus far, e.g., by the exclusion of periapical radiographs with upper anterior and posterior teeth or by the inclusion of anterior teeth only [23,26]. To avoid the influence of data inconsistencies on the results of the trained CNN, Tsoromokos et al. [26] only considered periapical radiographs of the mandible and reported a data set with 446 radiographs. In addition, Alotaibi et al. [23] considered 1724 periapical radiographs of maxillary and mandibular anterior teeth only and excluded radiographs of teeth that had been restored with full crowns or root canal treatments, as well as radiographs of teeth that had undergone apical surgery with root resection. In this context, the study by Lee et al. [28] should also be mentioned, which included periapical radiographs of posterior teeth to identify periodontally compromised premolars and molars. Further exclusion criteria were root canal treatment and teeth with fully restorative crowns as well as moderate to severe caries and teeth with a shape deviating from the usual anatomical structure. When considering the data shown in Table 4, it must be concluded that the partial exclusion of periapical radiographs may bias the model’s performance and limit the generalisability of the data shown. As is reasonable for this finding, the anatomical structures in the upper jaw in relation to the intraoral projection technique must be considered. Interestingly, this issue can be obviously downsized when using panoramic X-rays [15]. Nevertheless, a well-balanced inclusion of periapical radiographs from different groups of teeth may be relevant and should be implemented in future studies.

In this study, five well-established open-source transformer networks were trained: ViT-base and ViT-large from Google, BEiT-base and BEiT-large from Microsoft, and DeiT-base from Facebook/Meta [33,37,38]. The main differences between the transformer networks are in their size, training strategy and fine-tuning approach. “Base” and “large” models differ in size and computational complexity, whereby “large” models have more parameters. During training, ViTs process images as a sequence of patches and use an attention mechanism to learn the overall correlations within images. DeiT can achieve a high performance even with limited training data. Here, a smaller model learns to imitate a larger, already pre-trained model and benefits from a large data set without directly using it. In contrast, BEiT is trained in a two-stage process: pre-training on a large data set to capture general visual features, followed by fine-tuning for specific tasks. Transformer networks have rarely been applied for computer vision tasks in dentistry and not specifically for the detection of PBL. So far, only three studies using transformer networks were published; however, none of them focused on PBL assessment in periapical radiographs [50,51,52]. Nevertheless, there have been studies in which CNNs were used for PBL detection on periapical and panoramic radiographs (e.g., [11,14,15,17,21,22,23,24,25,26,40]). Here, the majority of investigations used only a low to moderate number of radiographs for model development, and most studies on periapical radiographs included a maximum of a few thousand images [13,22,23,25,27,28]. In contrast, Kim et al. [20] annotated the PBL in an extensive set of 12,179 panoramic radiographs, which may have potentially enhanced the internal study strength. The reported model-dependent AUC values ranged from 0.92 to 0.95 [20], which were slightly higher than the results from our study setup (Table 2). Therefore, it can be argued that the chosen study setup produced comparable data in the moment, which in part might be attributed to the use of transformer networks. Interestingly, we observed similar performance data with each of the included transformer networks. There was a minor tendency for less-complex transformer networks, e.g., Google’s vision transformer/base, to perform better than their more complex counterparts (Table 2 and Table 3). However, further improvements might be possible, especially by employing exact annotations in a large image set. Such features could enable precise object segmentation [20].

This study has several strengths and limitations. From a methodological point of view, this study used a large and well-balanced set of periapical radiographs (*N* = 21,819) in which all X-rays were diagnosed by dental professionals following the latest recommendations for PBL assessment [7,8]. Another unique feature seems to be the comparison of five transformer networks for the detection of PBL on periapical radiographs, as no other studies with the same methodology could be identified. In addition, the following limitations must be taken into account. In this study, we used categorial diagnostic scoring per image only. In detail, this means that the exact areas of PBL on periapical radiographs remained unmarked, which can be interpreted as a limitation. The exact annotation must be understood as a crucial feature to localize PBL precisely on X-rays. The exact annotation of the pathological structures would require the detection, classification and segmentation of PBL on each radiographic image. In particular, the marking of pathological segments must be understood as a time-consuming procedure that needs to be addressed in future projects. Another limitation is that only periapical radiographs were examined in this study and that panoramic radiographs have not been considered so far. However, in view of the fact that both types of radiographs are commonly used to assess PBL, but the format, size and radiographic anatomy differ, a separate analysis was justified. In addition, no clinical information was available for the anonymized radiographs in this study. Another limitation might be that we did not include any other transformer networks or CNNs in this study.

## 5. Conclusions

From the results of this study, it can be concluded that it was possible to achieve good diagnostic performance for automatized PBL detection when using a large set of periapical radiographs and several transformer networks. However, it can be hypothesized that the model performance can be improved by using exact annotations.

## Figures and Tables

**Figure 1 diagnostics-13-03562-f001:**
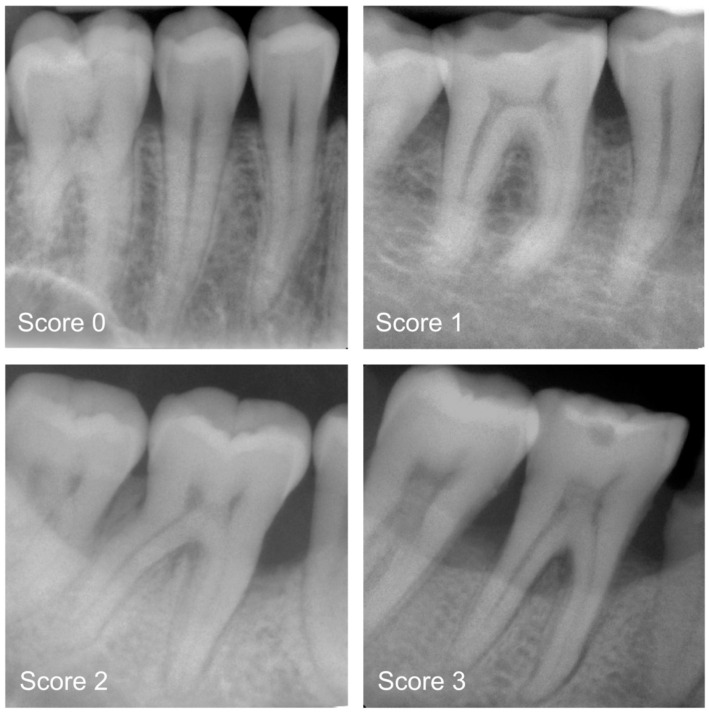
Examples of periapical radiographs for all categories: healthy periodontium (Score 0), mild radiographic periodontal bone loss (PBL) up to 15% of the root length (Score 1), moderate radiographic PBL between 15% and 33% of the root length (Score 2), and severe radiographic PBL extending to the mid–third of the root and beyond (Score 3).

**Figure 2 diagnostics-13-03562-f002:**
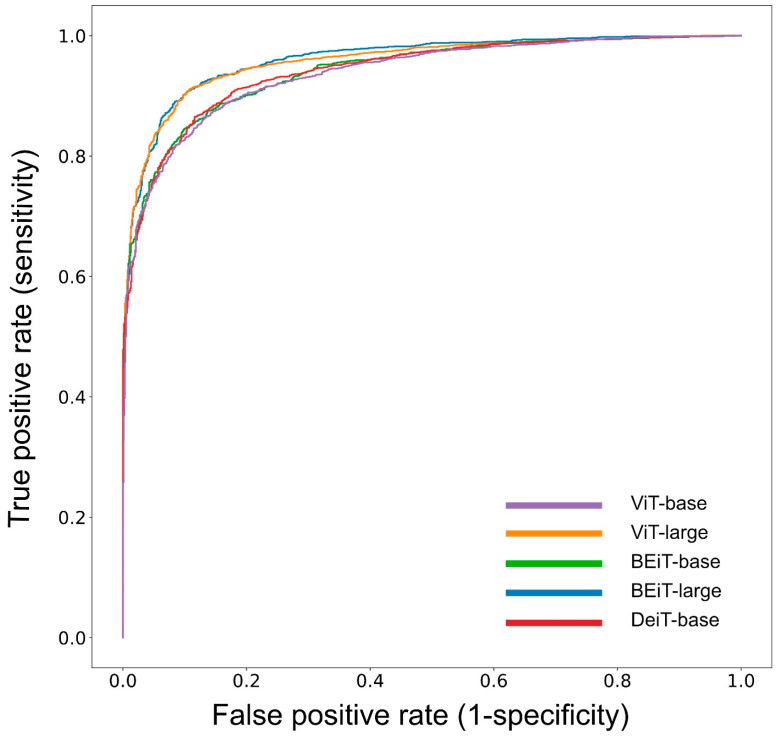
The receiver operating characteristic (ROC) curves illustrate the diagnostic performance of five different transformer networks for PBL detection.

**Table 1 diagnostics-13-03562-t001:** Overview of the included periapical radiographs (*N* = 21,819) in relation to the corresponding regions and categories of periodontal bone loss.

Region of Periapical Radiograph	Healthy Periodontium (Score 0)	Mild PBL (Score 1)	Moderate PBL (Score 2)	Severe PBL (Score 3)	Total (*N*)
1st Quadrant	1701 (35.8%)	1826 (38.5%)	851 (18.0%)	367 (7.7%)	4745
2nd Quadrant	1231 (26.1%)	2080 (44.1%)	1093 (23.2%)	312 (6.6%)	4716
3rd Quadrant	1477 (34.7%)	2033 (47.7%)	593 (13.9%)	157 (3.7%)	4260
4th Quadrant	1282 (30.8%)	2027 (48.7%)	713 (17.1%)	143 (3.4%)	4165
Maxillary anteriors	653 (33.6%)	661 (34.0%)	433 (22.3%)	197 (10.1%)	1944
Mandibular anteriors	202 (10.2%)	676 (34.0%)	786 (39.5%)	325 (16.3%)	1989

**Table 2 diagnostics-13-03562-t002:** Model characteristics of the used transformer networks.

	ViT-Base (Google)	ViT-Large (Google)	BEiT-Base (Microsoft)	BEiT-Large (Microsoft)	DeiT-Base (Facebook/Meta)
Neural network	Vision transformer		Bidirectional encoder representation from image transformers		Data-efficient image transformer
Epochs	5	5	5	5	5
Learning rate	0.00005	0.00005	0.00005	0.00005	0.00005
FLOS	7.280 × 10^15^	25.735 × 10^15^	7.277 × 10^15^	25.744 × 10^15^	7.280 × 10^15^
Samples per second	298.6	111.7	274.4	102.9	298.5
Parameter count	85.8 × 10^6^	303.3 × 10^6^	85.7 × 10^6^	303.4 × 10^6^	85.8 × 10^6^

**Table 3 diagnostics-13-03562-t003:** Overview of the overall diagnostic performance of the five transformer neuronal networks where the independent test set (*N* = 3000 radiographs) was evaluated by the AI-based algorithm for the assessment of periodontal bone loss. Diagnostic accuracy (ACC), sensitivity (SE), specificity (SP), negative predictive value (NPV), positive predictive value (PPV) and area under the receiver operating characteristic curve (AUC) were calculated for all types of teeth.

All Apical Radiographs	True Positive (TP)		True Negative (TN)		False Positive (FP)		False Negative (FN)		Diagnostic Performance					
	*N*	%	*N*	%	*N*	%	*N*	%	ACC	SE	SP	NPV	PPV	AUC
ViT-base	1884	62.8	673	22.4	230	7.7	213	7.1	85.2	89.8	74.5	76.0	89.1	0.918
ViT-large	1831	61.0	671	22.4	232	7.7	266	8.9	83.4	87.3	74.3	71.6	88.8	0.899
BEiT-base	1885	62.8	649	21.6	254	8.5	212	7.1	84.5	89.9	71.9	75.4	88.1	0.914
BEiT-large	1914	63.8	631	21.0	272	9.1	183	6.1	84.8	91.3	69.9	77.5	87.6	0.907
DeiT-base	1879	62.6	646	21.5	257	8.6	218	7.3	84.2	89.6	71.5	74.8	88.0	0.908

**Table 4 diagnostics-13-03562-t004:** Overview of the diagnostic performance of the five transformer neuronal networks for mandibular and maxillary anterior and posterior teeth. Accuracy (ACC), sensitivity (SE), specificity (SP), negative predictive value (NPV), positive predictive value (PPV) and area under the receiver operating characteristic curve (AUC) were calculated.

		True Positive (TP)		True Negative (TN)		False Positive (FP)		False Negative (FN)		Diagnostic Performance					
		*N*	%	*N*	%	*N*	%	*N*	%	ACC	SE	SP	NPV	PPV	AUC
**Mandibular anterior teeth**	ViT-base	240	88.2	16	5.9	9	3.3	7	2.6	94.1	97.2	64.0	69.6	96.4	0.944
	ViT-large	241	88.6	18	6.6	7	2.6	6	2.2	95.2	97.6	72.0	75.0	97.2	0.960
	BEiT-base	242	89.0	21	7.7	4	1.5	5	1.8	96.7	98.0	84.0	80.8	98.4	0.963
	BEiT-large	245	90.1	18	6.6	7	2.6	2	0.7	96.7	99.2	72.0	90.0	97.2	0.952
	DeiT-base	242	89.0	15	5.5	10	3.7	5	1.8	94.5	98.0	60.0	75.0	96.0	0.970
**Mandibular posterior teeth**	ViT-base	700	61.6	287	25.3	78	6.9	70	6.2	87.0	90.9	78.6	80.4	90.0	0.937
	ViT-large	687	60.5	285	25.1	80	7.1	83	7.3	85.6	89.2	78.1	77.4	89.6	0.913
	BEiT-base	704	62.0	277	24.4	88	7.8	66	5.8	86.4	91.4	75.9	80.8	88.9	0.933
	BEiT-large	711	62.6	279	24.6	86	7.6	59	5.2	87.2	92.3	76.4	82.5	89.2	0.923
	DeiT-base	694	61.1	281	24.8	84	7.4	76	6.7	85.9	90.1	77.0	78.7	89.2	0.927
**Maxillary** **anterior teeth**	ViT-base	157	59.5	81	30.7	18	6.8	8	3.0	90.2	95.2	81.8	91.0	89.7	0.958
	ViT-large	156	59.1	77	29.2	22	8.3	9	3.4	88.3	94.5	77.8	89.5	87.6	0.948
	BEiT-base	158	59.8	73	27.7	26	9.8	7	2.7	87.5	95.8	73.7	91.3	85.9	0.954
	BEiT-large	157	59.5	73	27.7	26	9.8	8	3.0	87.1	95.2	73.7	90.1	85.8	0.954
	DeiT-base	154	58.3	75	28.4	24	9.1	11	4.2	86.7	93.3	75.8	87.2	86.5	0.954
**Maxillary posterior teeth**	ViT-base	787	59.2	289	21.8	125	9.4	128	9.6	81.0	86.0	69.8	69.3	86.3	0.875
	ViT-large	747	56.2	291	21.9	123	9.3	168	12.6	78.1	81.6	70.3	63.4	85.9	0.851
	BEiT-base	781	58.8	278	20.9	136	10.2	134	10.1	79.7	85.4	67.1	67.5	85.2	0.865
	BEiT-large	801	60.3	261	19.6	153	11.5	114	8.6	79.9	87.5	63.0	69.6	84.0	0.861
	DeiT-base	789	59.4	275	20.7	139	10.4	126	9.5	80.1	86.2	66.4	68.6	85.0	0.860

## Data Availability

The data that support the findings of this study are available from the corresponding author upon reasonable request.

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
