# Peer review of "Automatized Detection of Periodontal Bone Loss on Periapical Radiographs by Vision Transformer Networks"

_diagnostics, 2023, doi:10.3390/diagnostics13233562_

Round 1
Reviewer 1 Report
Comments and Suggestions for Authors
Dear authors,
the article is well written but needs some modification in some area. Such as
1. In introduction, need to highlight more on the algorithms. Various articles need to be discussed in introduction which supports the need for the study.
2. In Methodology, inclusion criteria need to be elaborated.
3. Discussion, more studies can be discussed regarding the algorithms or transformer networks before discussing.
Comments on the Quality of English Language
Minor English correction required.
Author Response
Thank you for your feedback. Please see the attachment.

Reviewer 2 Report
Comments and Suggestions for Authors
The study is extremely interesting, and overall well conducted. I would only suggest the authors to clearly state that, although the technique employed may be valuable in the assessment of periodontal bone loss, the diagnosis of periodontal disease should always be made clinically as radiographs tend to underestimate the extent and activity of periodontitis (see 10.1111/j.1600-051x.1977.tb01879.x). Apart from this, the study methodology is adequate and results properly reported.
Author Response

(The authors gave the same response as above.)

Reviewer 3 Report
Comments and Suggestions for Authors
Dear authors!
Artificial intelligence (AI) is playing an increasingly important role in dentistry, offering new opportunities and advantages in diagnosis, treatment planning and the treatment process. With the help of machine learning algorithms and image processing, AI can help in visualization and analyzes of teeth and gums images, such as X-rays, to aid in the diagnosis of cavities, periodontitis, and other dental problems. This can help dentists to determine the pathology without bias, assess its degree and choose the most effective treatment strategy. Nowadays, there are many transformer networks that can be used for these purposes, but the reliability and accuracy of these techniques is still in question. Thus, the presented study, which is aimed at comparing the diagnostic characteristics of transformer networks for automatized detection of periodontal bone loss, is relevant. It is worth noting that the study is original, since previously published articles, indeed, have not considered the presented in the study transformer networks for periodontal bone loss detection.
The design of the study is simple and well understood, and also corresponds to the purpose of the study. The manuscript has a clear structure and is easy to read. The "Materials and methods" section is described in sufficient detail to ensure the reproducibility of the research and complies with the necessary requirements, including ethical ones. The results of the study are presented in detail both in the text and in tables. However, it is worth paying attention to the duplicate heading of the "Table 4" on lines [155-159] and [161-164] and exclude the extra text. The conclusions are formulated correctly and correspond to the aim of the study. Literary references are relevant as most of them have been published no earlier than 10 years. Excessive number of self-citations is excluded.
In general, the article leaves a good impression, thus, I can recommend this article for publication.
Author Response

(The authors gave the same response as above.)
